# The Landscape of lncRNAs in Multiple Myeloma: Implications in the “Hallmarks of Cancer”, Clinical Perspectives and Therapeutic Opportunities

**DOI:** 10.3390/cancers14081963

**Published:** 2022-04-13

**Authors:** Ilaria Saltarella, Benedetta Apollonio, Aurelia Lamanuzzi, Vanessa Desantis, Maria Addolorata Mariggiò, Jean-François Desaphy, Angelo Vacca, Maria Antonia Frassanito

**Affiliations:** 1Department of Biomedical Sciences and Human Oncology, Unit of Internal Medicine “Guido Baccelli”, University of Bari Medical School, Piazza Giulio Cesare 11, I-70124 Bari, Italy; ilaria.saltarella@uniba.it (I.S.); benedettaapollonio@gmail.com (B.A.); aurelia.lamanuzzi@uniba.it (A.L.); vanessa.desantis@uniba.it (V.D.); angelo.vacca@uniba.it (A.V.); 2Department of Biomedical Sciences and Human Oncology, Pharmacology Section, University of Bari Medical School, Piazza Giulio Cesare 11, I-70124 Bari, Italy; jeanfrancois.desaphy@uniba.it; 3Department of Biomedical Sciences and Human Oncology, Unit of General Pathology, University of Bari Aldo Moro, I-70124 Bari, Italy; mariaaddolorata.mariggio@uniba.it

**Keywords:** multiple myeloma, hallmarks of cancer, lncRNA, prognosis, therapy

## Abstract

**Simple Summary:**

Multiple myeloma (MM) is an aggressive hematological neoplasia caused by the uncontrolled proliferation of aberrant plasmacells. Neoplastic transformation and progression are driven by a number of biological processes, called ‘hallmarks of cancer’, which are regulated by different molecules, including long non-coding RNAs. A deeper understanding of the mechanisms that regulate MM development and progression will help to improve patients stratification and management, and promote the identification of new therapeutic targets.

**Abstract:**

Long non-coding RNAs (lncRNAs) are transcripts longer than 200 nucleotides that are not translated into proteins. Nowadays, lncRNAs are gaining importance as key regulators of gene expression and, consequently, of several biological functions in physiological and pathological conditions, including cancer. Here, we point out the role of lncRNAs in the pathogenesis of multiple myeloma (MM). We focus on their ability to regulate the biological processes identified as “hallmarks of cancer” that enable malignant cell transformation, early tumor onset and progression. The aberrant expression of lncRNAs in MM suggests their potential use as clinical biomarkers for diagnosis, patient stratification, and clinical management. Moreover, they represent ideal candidates for therapeutic targeting.

## 1. Introduction

Multiple myeloma (MM) is a hematological neoplasia characterized by the aberrant growth of malignant plasma cells, also known as MM cells, in the bone marrow (BM) [1]. MM arises from the pre-neoplastic forms of the monoclonal gammopathy of undetermined significance (MGUS) with the 5% of risk to progress into MM in 5 years, and the smoldering myeloma (SMM) with 50% of progression risk [1]. The progression is strongly dependent on BM microenvironment that generates a permissive niche promoting MM cells viability via cytokines, growth factors, angiogenesis, aberrant cell-to-cell communications, deregulation of intracellular pathways and gene expression [2,3,4]. Alteration of transcription profile involves both coding genes, i.e., the activation of oncogenes or the inactivation of tumor suppressor genes [5], and a non-coding portion of transcriptome that constitutes almost the 95% of the human genome [6].

Non-coding RNAs (ncRNAs) were first considered as junk nucleic acids with no apparent protein-coding role [7]. In the last few years, long ncRNAs (lncRNAs) have become increasingly important as key regulators of gene expression and downstream biological functions, due to their multiple mechanisms of action. LncRNAs are transcripts longer than 200 nucleotides that are not translated into proteins. They regulate transcription, translation, and post-translational modifications by interacting with DNA, RNAs and proteins. They have also emerged as key drivers of many biological processes that enable cell malignant transformation, early tumor onset and progression known as “hallmarks of cancers [8,9]. Next-generation sequencing as well as microarray analysis have identified recurrent deregulated lncRNAs (e.g., MALAT1, H19, HOTAIR, ANRIL) across different cancer types that affect tumor-driving processes including proliferation, survival, migration or genomic stability [10,11,12,13,14]. Recently, Carlevaro-Fita et al. [15] generated the first Cancer LncRNA Census, a compendium of lncRNAs with a direct role in tumorigenesis, establishing a functional link between altered lncRNA profiles and cancer.

Here, we review the contribution of lncRNAs in the “hallmarks of cancer” in MM, focusing on their ability to regulate different intracellular pathways involved in the multi-step process of cancer development. We also analyze their putative role as clinical biomarkers for diagnosis, patient stratification, and management. Finally, we discuss their involvement in resistance to therapy and their potential use as therapeutic targets alone or in combination with the canonical anti-MM drugs.

## 2. LncRNAs Biogenesis and Functions

The majority of lncRNAs are transcribed by RNA Polymerase II from several genome loci leading to different classes that are grouped based on their orientation and localization in the genome as: (*i*) sense lncRNAs, transcribed in the sense of coding genes (5′-3′); (*ii*) anti-sense lncRNAs transcribed in the opposite strand of coding genes (3′-5′); (*iii*) intergenic lncRNAs (lincRNAs), located between two genes; (*iv*) intronic lncRNAs, entirely located in introns of coding genes; (*v*) enhancer lncRNAs (eRNA); (*vi*) promoter-associated lncRNAs; and (*vii*) untranslated region (UTR)-associated lncRNAs (Figure 1) [16].

After transcription, lncRNAs undergo alternative splicing that generates transcripts with few exons, but longer compared to mature mRNAs [16]. Although lncRNAs have been identified as ncRNAs, RNA-sequencing and mass spectrometry analyses have recently described the presence of a small open reading frame (sORF ≤ 300 nucleotides), suggesting that lncRNAs may encode for peptides/small proteins with regulatory functions [17]. Most lncRNAs have a 7-methyl guanosine (m^7^G) 5′ capping and a 3′ poly-A tail that preserves their integrity. lncRNAs lacking the poly-A tails are instead stabilized by the formation of secondary structures (i.e., triple-helical structures), and/or by the binding to small proteins that regulate their sub-cellular localization. Indeed, lncRNAs can be retained inside the nucleus, exported into the cytoplasm, compartmentalized in cellular organelles (i.e., mitochondria, endoplasmic reticulum, ribosomes), or packaged into extracellular vesicles (EVs) and delivered to neighboring cells [18]. Interestingly, the different cellular localization of lncRNAs influences their activity and biological functions. Nuclear lncRNAs are often involved in the modulation of chromatin and in the organization of the nucleus; cytosol-exported lncRNAs regulate gene expression at post-transcriptional level; mitochondria-associated lncRNAs regulate mitochondrial transcriptome; and ribosome-associated lncRNAs interact with the translational machinery.

Nuclear lncRNAs modulate epigenetic organization of the genome by sequestering chromatin modifiers, by recruiting them to the promoter or regulatory sites of target genes (cis-regulation), or by binding to distant regulatory sites (trans-regulation) [19]. They are also involved in the organization of nucleolar structures by ensuring the maintenance of nucleolar domains, such as the nucleolus, the nuclear speckles and paraspeckles that mainly contain a single type of RNA or protein needed for nucleolar activity. Interestingly, the epigenetic inactivation of the X-chromosome that occurs during female cell development is regulated by a series of lncRNAs, namely X-inactive specific transcript (Xist) located in the X chromosome inactivation center (XIC). During embryogenesis of female mammals, the activation of Xist recruits chromatin modifying complexes, i.e., histone modifiers and DNA methylation enzymes, that lead to chromosome X inactivation [20].

LncRNAs also affect gene expression by acting at post-transcriptional level regulating alternative splicing, transcription, and translation [21]. They directly bind to target mRNA or deliver RNA-binding proteins (RBPs) to regulate mRNA splicing, stability, and translation. Many lncRNAs act as decoys for miRNAs through the expression of specific miRNA-binding sites that operate as competitive endogenous RNAs (ceRNAs). This mechanism, known as “miRNA sponging”, modulates miRNA expression and affects the transcription of target genes [22].

Overall, based on the above described mechanisms of action, lncRNAs can also be classified in: (*i*) signal lncRNAs that regulate gene transcription acting alone or with other proteins (i.e., transcription factors); (*ii*) decoy lncRNAs that bind to specific protein, molecules or enzymes (i.e., chromosome folding proteins or transcription regulators) modulating their activity; (*iii*) guide lncRNAs that enable the recruitment of specific protein to their target genome locus to exert their functions (i.e., the cis- and trans- regulation); (*iv*) scaffold lncRNAs that facilitate the interaction of different molecules and protein allowing the assembly and the activity of macromolecular complexes (i.e., Xist lncRNA) [23].

Therefore, based on the multiple functions of lncRNAs in the remodeling of cell transcriptome, they are considered an emerging feature of cancer pathogenesis.

## 3. LncRNAs and “Hallmarks of Cancer” in MM

MM onset and progression involve a progressive transformation of both neoplastic and stromal cells. Hanahan and Weinberg [9] named the biological processes that contribute to tumor development as “hallmarks of cancer”. During the 22 years after their first conceptualization [9], the “hallmarks of cancer” have been redefined and expanded on several occasions, and nowadays they count a total of 14 different key signatures [24,25]. Based on the increasing importance of lncRNAs in MM pathogenesis, here we review their role as regulators of the “hallmarks of cancer” (Figure 2) [14].

### 3.1. Genomic Instability, Mutation, and Non-Mutational Epigenetic Reprogramming

MM progression and drug resistance are strongly influenced by genomic instability that increases the occurrence of genome mutations during the multi-step transformation process. Genomic instability causes several chromosomal alterations, i.e., gene deletion, duplication, microsatellite instability, structural variations, and mutations in oncogenes and tumor suppressor genes [26]. LncRNAs can contribute to genomic instability by modifying cell transcriptome and by inducing chromatin rearrangements and epigenetic regulation [6].

Hu et al. [27] demonstrated that plasma cells from MGUS and MM patients overexpress the lncRNA MALAT1. This lncRNA binds to PARP1/LIG3 complex that recognizes the double strand breaks on DNA and activates the DNA repair via the alternative non-homologous end-joining (A-NHEJ) pathway. Thus, MALAT1 overexpression triggers DNA repair with consequent induction of chromosome rearrangements and mutations. Accordingly, inhibition of MALAT1 by gapmer anti-sense oligonucleotides inhibits DNA repair and induces cell apoptosis both in vitro and in vivo [27].

Recently, the oncogenic lncNEAT1 has been demonstrated to regulate the DNA repair pathways. It controls gene expression, mRNA splicing and editing in response to cellular stress [28]. Gene expression profiling and gene set enrichment analysis (GSEA) of NEAT1-knockdown cells identified several genes involved in DNA repair, including the DNA-damage sensor protein RPA32, and the effector protein of DNA repair mechanisms, BRCA1. NEAT1 inhibition increases H2A.X expression, as a sign of DNA damage, and affects cell viability.

Analysis of lncRNAs profile in MM patients identified four lncRNAs (RP4-803 J11.2, RP1-43E13.2, RP11-553 L6.5, and ZFY-AS1) that significantly correlate to overall survival (OS) and have a prognostic significance. Functional analysis of mRNAs/lncRNAs co-expression predicts lncRNA activities and identifies six functional clusters operating in chromatin modification, DNA replication, DNA repair and RNA processing, suggesting that these lncRNAs are involved in genetic and epigenetic events that occur during MM progression [29].

Finally, other studies documented that some lncRNAs (such as PVT1, MALAT1, ANRIL, GUARDIN, NEAT1, MEG3, PANDA) are involved in the regulation of the “genome guardian” p53. Deregulation of p53-associated lncRNAs may support the survival of MM cells with damaged DNA [30].

### 3.2. Sustaining Proliferative Signaling, Evading Growth Suppressors and Resisting Cell Death 

During progression, MM cells increase their proliferation and resistance to apoptosis through several factors that inhibit growth suppressors and sustain proliferative pathways. Based on the pleiotropic functions of lncRNAs and on their aberrant expression in MM [31], the correlation among lncRNAs, cell proliferation and resistance to apoptosis has been deeply studied. Aberrant expression of MALAT1 has been described in different cells from BM microenvironment and correlates to patients’ outcome [32]. Liu et al. [33] demonstrated a direct correlation between MALAT1 expression and MM cells proliferation as well as apoptosis resistance. Transient transfection of MM cell lines with anti-MALAT1 short hairpin RNAs (shRNAs) halts cell cycle in G1 phase, induces apoptosis by modulating the pro-apoptotic Bax and the anti-apoptotic Bcl-2 proteins in vitro, and reduces tumor growth in xenografted mice [33]. Moreover, MALAT1 regulates MM cell viability and apoptosis through autophagy by increasing high mobility group box 1 (HMGB1) expression [34]. MALAT1-induced autophagy reduces the anti-MM effect of bortezomib [34], indicating that MALAT1 targeting may prevent the activation of autophagy induced by bortezomib [35].

LncRNA PCAT-1 is also overexpressed in MM and promotes tumor cell survival [36]. Gene ontology and KEGG pathway analysis as well as molecular biology studies demonstrated that PCAT-1 activates the JNK/MAPK pathways [36] that sustain MM cell survival [3]. Furthermore, Liu et al. [37] demonstrated that the lncRNA LUCAT1 promotes MM cell proliferation, preventing cell cycle arrest in S phase and apoptosis via Smad2 phosphorylation and TGF-β signaling activation [37].

Additional studies described the oncogenic role of lncCRNDE. The CRISPR-selective deletion of the CRNDE reduces MM cells proliferation, adhesion, and tumor growth by downregulating the expression of IL-6R, hence preventing the activation of IL-6/IL-6R pathway [38], essential for MM pathogenesis and progression [39]. CRNDE also affects MM cell proliferation and apoptosis by acting as ceRNA for miR-451 [40], indicating that a single lncRNA may act through several mechanisms.

Other lncRNAs play a tumorigenic role by regulating miRNAs expression through their sponge activity. Yang et al. [41] showed that lncPVT1 is a sponge for miR-203a and its overexpression supports tumor growth. TUG1 is involved in the miR-34a-5p/NOTCH1 axis and affects cell proliferation and apoptosis by regulating Hes-1, Survivin, and Bcl-2 expression [42]. Conversely, the tumor suppressor MEG3 is downregulated in MM. Its overexpression reduces miR-181a and increases the levels of its mRNA target, HOXA11, reducing cell proliferation and apoptosis resistance [43]. Recently, we have demonstrated that HOTAIR, TOB1-AS1, and MALAT1 are decoys for miR-23b-3p, miR-27b-3p, and miR-125b-5p. These lncRNAs sponge miRNAs with a “tumor suppressor” function transferred by fibroblast-derived EVs into recipient MM cells. These results suggest that MM cells actively discriminate miRNAs expression via lncRNAs and neutralize exosomal miRNAs to ensure their survival [44]. Deng et al. [45] showed that mesenchymal stromal cells (MSC)-derived EVs promote MM progression by delivering the lncRNA LINC00461. This lncRNA is a ceRNA of miR-15a/miR-16 and increases the levels of the Bcl-2 protein, demonstrating that its upregulation induces cell proliferation and prevents cell apoptosis.

Overall, these data support the hypothesis that lncRNAs are new players in the regulation of cell transcriptome and intracellular pathways with a pivotal role in tumor cell growth and evasion of cell apoptosis (Figure 2).

### 3.3. Inducing or Accessing Vasculature

Angiogenesis contributes to MM progression by providing nutrients, oxygen, and growth factors [46]. The transition from a physiological and controlled angiogenesis to a pathological and aberrant overangiogenic state is referred as “angiogenic switch”. During this phase, tumor cells support angiogenesis by inducing the aberrant expression of pro-angiogenic factors, ncRNAs and activation of intracellular pathways in the tumor microenvironment. These events lead to enhanced angiogenesis, characterized by altered vessel structure with abnormal flow and hyperpermeability [4].

Several studies documented the emerging role of lncRNAs in the regulation of angiogenesis and vascular disease identifying specific “angio-lncRNAs”, i.e., MALAT1, MANTIS, PUNISHER, MEG3, MIAT, SENCR and GATA6-AS [47]. In hepatocellular carcinoma, lncOIP5-AS1 acts as a ceRNA of miR-3163 and contributes to cell migration and angiogenesis by increasing VEGF-A levels [48]. Other lncRNAs regulate the VEGF levels via the modulation of the intracellular STAT3, PI3K/Akt/mTOR and Wnt/β-catenin pathways [49,50,51]. As solid tumors, hematological malignancies, including MM, undergo the “angiogenic switch” [4,52]. The BM of MM patients with active disease shows an overangiogenic state compared to MGUS or healthy subjects, implying the importance of neovessel formation in supporting disease progression. MM cells as well as BM stromal cells, such as fibroblasts, immune, and endothelial cells, create a pro-angiogenic niche that supports angiogenesis via different mechanisms, which may represent an attractive therapeutic target [4,52,53].

Allegra et al. [53] analyzed lncRNAs involved in bone homeostasis and cancer development in lymphomonocytes of MM patients and healthy subjects. They observed that MEG3, MANTIS and HIF1A-AS2 were upregulated in MM and involved in angiogenesis. MEG3 knockdown restrains VEGFR2 expression, thus blocking the VEGF-dependent sprouting of human umbilical vein endothelial cells (HUVECs) [54]. Moreover, although different studies described a MEG3 downregulation in MM [43], Allegra et al. [53] have instead found increased levels of MEG3 in patients treated with bisphosphonates that develop osteonecrosis of the jaw (BONJ). MANTIS is an epigenetically regulated lncRNA overexpressed in HUVECs. It controls several angiogenesis-related mRNAs (SOX18, SMAD6, and COUP-TFII) and ensures endothelial cells sprouting and migration [55]. HIF1A-AS2 acts as sponge of miR-153-3p, and promotes the expression of HIF-1α, VEGFA and Notch1. Activation of HIF-1α/VEGFA/Notch1 pathway enhances HUVECs viability, migration, and tube formation [56]. During MM progression, hypoxia induces H19 overexpression, which modulates HIF-1α targets, i.e., VEGF, C-X-C chemokine receptor type 4 (CXCR4) and the transcription factors Snail and Slug. Inhibition of H19 reduces HIF-1α activation and, consequently, the expression of HIF-1α targets, thus affecting the adhesion of MM cells to the BM stromal cells [57].

Overall, the pro-angiogenic role of lncRNAs in MM is indirectly supported by literature data demonstrating the importance of VEGF/VEGFR2 axis [58], HIF-1α [59] and Notch1 [60,61] pathways in BM angiogenesis.

### 3.4. Tumor-Promoting Inflammation and Avoiding Immune Destruction

MM progression and resistance to therapy are associated to immune dysfunction, caused by an altered immune milieu and by inactivation or tolerization of effector immune cells [62]. The mechanisms responsible for the creation of an immunosuppressive microenvironment in solid and hematological malignancies have been extensively characterized and require the presence of a constant bidirectional crosstalk between tumor and surrounding immune and accessory cells [63].

Recent data have demonstrated that lncRNAs are involved in the regulation of adaptive and innate immune responses [64]. Specific lncRNAs, expressed both by cancer and immune cells, can regulate the function and the composition of the immune infiltrate in the tumor microenvironment. For example, by sponging miR-195 and miR-5590-3p, respectively, MALAT1 and SNHG14 induce PD-L1 upregulation in lymphoma cells that inhibit the cytolytic activity of CD8^+^ T lymphocytes [65,66]. The upregulation of different lncRNAs (GNAS-AS1, XIST, P21, ANCR, and MM2P) induces M2 polarization of tumor-associated macrophages in solid cancers (reviewed in [67]). Moreover, different lncRNA profiles have been correlated to the immune infiltrate, OS, and response to immunotherapy in bladder tumors, melanoma, and breast cancer [68,69].

Until now, only a few reports have analyzed the effect of lncRNAs in the regulation of MM immunity, and no study has shown a direct alteration of lncRNA expression in immune cells.

Patients with high levels of the lncRNA NR_046683 in BM samples display short progression free survival (PFS) and significantly higher levels of β2-microglobulin, indicating that this lncRNA can be involved in the progression of MM. Interestingly, the analysis of lncRNA-mRNA network predicted a correlation between NR_046683 and the genes involved in leukocytes activation and immune responses, suggesting that it could modulate multiple aspects of immune regulation [70]. However, these data still lack functional demonstration.

Gao et al. have demonstrated that NEAT1 can indirectly affect macrophage activation [71]. A screening analysis revealed that NEAT1 sponges miR-214, favoring the upregulation of B7-H3 on MM cells [71]. The transmembrane protein B7-H3 is a dual immune regulator, acting both as an inhibitory ligand for tumor infiltrating CD8 T lymphocytes and as an activation signal for M2 macrophages through the JAK/STAT signaling [72]. MM cells treated with a shNEAT1 showed decreased levels of B7-H3 and a reduced ability to polarize the pro-angiogenic, immunosuppressive M2 macrophages [71]. These data offer the first demonstration that altered lncRNA profiles in tumor cells govern the shaping of the immune microenvironment in MM (Figure 2).

### 3.5. Activating Invasion and Metastasis: The Osteolytic Bone Disease (OBD) 

OBD is a common clinical manifestation of MM and has a negative impact on patients’ quality of life and survival [73]. OBD is caused by a disequilibrium between osteoblasts and osteoclasts, that, respectively, drive bone formation and resorption [74]. By invading the BM microenvironment, MM cells hijack both extracellular matrix and bone-resident accessory cells and foster the creation of a pro-tumoral niche, in which cytokines and adhesion molecules promote tumor cell survival and proliferation, ultimately leading to bone lesions [75]. The mechanisms underlying OBD are complex and probably linked to altered lncRNA profiles. Allegra et al. [53] identified 15 lncRNAs differentially expressed in MM patients treated with bisphosphonates that developed BONJ. The identified lncRNAs were predicted to target pathways involved in bone formation and metabolism. Among them, DANCR and MALAT1 were downregulated, while HOTAIR, MEG3 and H19 were upregulated in patients with BONJ compared to controls. Although these lncRNAs regulate bone remodeling in several conditions [76], their functional contribution in the MM OBD has not been studied yet.

Other lncRNAs have been functionally linked to the increased migration and invasion of MM cells, a key process in bone disruption. The lncRNA linc01606 is upregulated in circulating mononuclear cells of MM patients compared to controls and is associated to a shorter OS [77]. As shown in gastric and breast cancer, linc01606 expression fosters proliferation, migration, and invasion of MM cells, potentially by sponging miR579-3p [77,78]. BM742401, a known tumor suppressor lncRNA found in chronic lymphocytic leukemia [79], is instead epigenetically silenced in both MGUS and MM patients, indicating that this could be an early event in MM onset. Re-expression of BM742401 using a demethylating agent (5-AzadC) decreases MM cell migration, and possibly reduces tumor cell infiltration in the bone [26].

Bone disease is also associated to the imbalanced activity of MSCs, which are the primary source of osteoblasts. MM cells suppress the osteoblastic potential of MSCs through the release of soluble molecules and the establishment of adhesive contacts [75]. In addition, MM can hijack MSCs through the secretion of EVs [80], whose cargo includes lncRNAs [81]. Specifically, RUNX2-AS1, an antisense transcript arising from intron 7 of the RUNX2 gene, can be transferred to MSCs via MM-derived EVs. Once released in the MSCs cytoplasm, RUNX2-AS1 directly binds to RUNX2 pre-mRNA, interferes with its correct splicing, and causes RUNX2 protein downregulation. RUNX2 is a transcription factor involved in the initiation of osteoclastogenesis and its decreased expression reduces the osteogenic potential of MM MSCs [82,83].

Other reports have shown that MM MSCs display an altered lncRNA expression profile. HOXC-AS3, a natural antisense transcript of HOXC10, is upregulated in MM MSCs. It positively regulates the expression of HOXC10 by binding and stabilizing HOXC10 mRNA. Enhanced mRNA stability is associated to upregulated HOXC10 expression and decreased osteogenic differentiation of MSCs both in vitro and in vivo [84].

MALAT1 is overexpressed in MM MSCs where it acts as a transcriptional co-activator of the neighboring gene LTBP3. Specifically, MALAT1 forms a complex with the transcription factor Sp1, stabilizing its binding to the LTBP3 promoter and activating LTBP3 transcription [85]. LTBP3 induces TGFβ1 secretion by MM MSCs, which fosters OBD [85,86,87] (Figure 2).

Finally, a recent study has demonstrated that lncRNAs are also altered in MM osteoblasts compared to healthy controls. The analysis by Peng and colleagues [88] has revealed that MM osteoblasts display an upregulated expression of the new lncRNA LINC01473 together with CD74 mRNA. Although the functional consequences of this altered profile have not been investigated yet, authors suggested their possible involvement in OBD and immune escape [88].

### 3.6. Deregulating Cellular Metabolism

Tumor growth strongly depends on the reprogramming of cell metabolism to ensure cell proliferation, survival, and to avoid immune surveillance [89]. Warburg et al. [90] firstly described the ability of cancer cells to metabolize glucose into lactate even in the presence of oxygen. This mechanism is called aerobic glycolysis or “Warburg effect”. This metabolic deregulation also involves other metabolic pathways, such as the tricarboxylic acid cycle, the oxidative phosphorylation, amino acid and lipid metabolisms [91].

Oncogenic lncRNAs modulate several features of cancer cell metabolism. For example, MALAT1 regulates glucose metabolism through the miR-1271-5p/SRY-Box 13 (SOX13) axis [92]. Its inhibition decreases glucose consumption as well as lactate and ATP production by affecting the levels of the hexokinase HK2 and of the glucose transporter GLUT1.

Yang et al. [93] demonstrated that PDIA3P modulates MM cells metabolism via the pentose phosphate pathway. Nuclear magnetic resonance spectroscopy showed that PDIA3P overexpression increases the consumption of ^13^C-labeled glucose and the formation of ^13^C-labeled lactate, implying an increase of pentose phosphate flux. Authors also demonstrated that PDIA3P interacts with c-Myc and triggers its binding to glucose-6-phosphate dehydrogenase (G6PD) promoter leading to an increase of G6PD expression. Activation of pentose phosphate pathway sustains MM cells proliferation and drug resistance [93].

### 3.7. Enabling Replicative Immortality and Senescent Cells 

Malignant cells acquire an unlimited replicative potential that supports tumor growth bypassing the cell “expiration date”, preventing telomere shortening and cell death [25]. Telomere maintenance is ensured by different mechanisms, including lncRNAs. Although the role of lncRNAs in the regulation of telomere-length has been established in several solid tumors [94], their involvement in MM has not been investigated yet. In solid tumors, lncTERC acts as a template for DNA telomere sequence, TERT functions as a catalytic component, and the lncTERRA regulates telomere length [94].

The updated version of “hallmarks of cancer” introduced cellular senescence as a new mechanism promoting malignant onset by the release of senescence-associated factors that modify the surrounding microenvironment [95]. The involvement of lncRNAs in regulation of cell senescence, i.e., oxidative stress, DNA damage, p53 deregulation, and hypoxia [31] has been previously described throughout the review.

Overall, these data offer an emerging perspective for cancer research in the field of lncRNA and oncology (Figure 2).

### 3.8. Unlocking Phenotype Plasticity

To evade the antiproliferative barrier typical of normal tissues, cancer cells unlock their terminal differentiation programs, gaining features of less differentiated or progenitor cells [24]. In MM, tumor plasma cells with a mature phenotype gain a certain degree of phenotypic plasticity by expressing specific stem cell progenitor markers, such as SOX2, MAGE, CD117 (KIT), and Nestin [96,97,98,99]. Although no link has been established so far between the expression of stem cell markers and specific lncRNAs in MM, silencing of MALAT1 has been shown to reduce stemness of glioma cells [100], suggesting that this lncRNA could foster phenotypic plasticity also in MM.

### 3.9. Polymorphic Microbiomes

Microbioma, the collection of resident microorganisms living inside our body, establishes a symbiotic crosstalk with the host influencing different aspects of human health. Several studies have demonstrated that commensal bacteria can foster tumorigenesis through different mechanisms: (*i*) production of toxins that induce DNA damage in surrounding cells, (*ii*) stimulation of epithelial cell proliferation, and (*iii*) induction of an altered immune cells activation status [24].

Few reports have shown that microbioma composition can influence the lncRNA expression profiles of cells from different sites, such as liver, colon, and adipose tissue [101,102]. Commensal bacteria can also alter lncRNAs expression profiles in immune cells. For example, the butyrate produced by gut microbioma induces the expression of lncLy6C, responsible for the differentiation of Ly6C^high^ inflammatory macrophages into Ly6C^int/neg^ resident macrophages [103]. Even though recent findings have demonstrated the role of gut microbiota alterations in MM progression and response to therapy [104,105,106], additional studies are needed to determine the impact of gut microbioma on the MM lncRNA landscape.

## 4. Clinical Perspectives and Therapeutic Opportunities of lncRNA

### 4.1. LncRNAs and Patient Stratification

Considering their ability to regulate and influence the hallmarks of cancer, lncRNAs could provide a useful tool to predict patient’s prognosis (Table 1).

Increased expression of several lncRNAs, such as MALAT1 [32], NEAT1 [107,113], CRNDE [38], PVT1 [108], TUG [42], UCA1 [109], LINC00461 [45], and TCF7 [110], has been correlated to shorter PFS, and downregulated expression of MEG3 [43] and OIP5-AS1 [111] has been associated to worse prognosis. In addition, a link between lncRNAs expression and drug resistance has been defined both in solid tumors [114] and MM (reviewed in the paragraph below), supporting their potential use as treatment-decision-making tools.

The development of high-throughput screenings has expanded our knowledge on lncRNAs, and deep sequencing approaches have demonstrated that lncRNA patterns are progressively changing in healthy and tumor plasma cells of MGUS, SMM, MM and plasma cell leukemia (PCL) [115]. Different lncRNA expression signatures have been associated to specific genomic alterations with prognostic significance, such as t(11;14), t(4;14), MAF translocations or hyperdiploidy status [112], while other studies have used lncRNA profiles as independent prognostic markers for MM patients [10,29]. Zhou et al. [29] developed a lncRNA-focused risk model for survival prediction based on the expression of 4 lncRNAs (RP4-803 J11.2, RP1-43E13.2, ZFY-AS1 and RP11-553 L6.5) that regulate processes such as cell proliferation and RNA repair [29].

Despite the association of altered lncRNAs with prognosis, their complex mechanism of action provides a limited predictive potential, and the use of prognostic models that combine multiple factors would increase prediction accuracy. Two independent reports have demonstrated that multivariate analysis linking the expression of single lncRNAs or lncRNA profiles with clinical and genetic risk factors allows patient stratification, potentially improving patients’ management [10,116].

Moreover, since lncRNAs regulate gene expression at different levels (see “LncRNA and gene regulation” section), a robust predictive model should consider the complex networks between coding and non-coding RNAs [117]. Ronchetti et al. [118] have shown that in silico analysis can help predicting miRNA-lncRNA pairs with a pathogenic function in MM. For example, lncMCL1-2 expression is negatively correlated to its targets mir106a-5p, miR18a-5p, miR18b-5p and miR17-5p that modulate MCL1 expression, suggesting that the miRNA sponging activity of lncMCL1-2 can directly affect survival of MM cells [118]. Zhu et al. [119] have also associated mRNAs and lncRNAs expression profiles that correlate to disease prognosis. They identified 39 lncRNAs and 1445 mRNAs involved in several biological processes with a direct impact on MM clinical course [119]. In addition, a more recent predictive model has identified the correlations between different players of the cellular transcriptome (i.e., lncRNA, miRNA, circRNA, and mRNA) and pathogenetic processes in MM [120], indicating that combining transcriptomic networks (lncRNAs-miRNAs-mRNAs) could give helpful insights in the dissection of MM evolution providing useful tools for patients clinical management.

### 4.2. Resistance to Therapy

Despite the advent of new therapeutic targets and the recent advances in drug discovery, MM patients ultimately relapse and develop resistance to currently available anti-MM drugs [121]. Several studies investigated the molecular mechanisms that regulate drug resistance, i.e., intracellular pathways, cytokines, and lncRNAs [122]. Malek et al. [123] demonstrated a deregulation of lncRNAs in MM cells resistant to proteasome inhibitors (bortezomib, carfilzomib, or ixazomib). Genome-wide profiling identified seven lncRNAs (lncCol4A2-1, lncZNF726-4, lncDNAJB11-6, lncMYOT-1, lncPRKCQ-1, lncCXADR-1, and lncZNF99-6) upregulated in MM cell lines and CD138^+^ plasma cells resistant to proteasome inhibitors compared to sensitive cells, suggesting their involvement in drug resistance [123]. More recently, H19, PCAT1, ANRIL and MALAT1 were proposed as inducer of bortezomib resistance [27,36,124,125]. H19 sustains tumor growth and bortezomib resistance by inhibiting the tumor suppressor miR-29b-3p and by increasing the levels of the anti-apoptotic protein MCL-1. Accordingly, sera from bortezomib-resistant MM patients showed elevated levels of H19 and MCL-1 and low levels of miR-29b-3p [126]. PCAT1 is overexpressed in MM patients and promotes cell growth and drug resistance via p38 and JNK/MAPK pathways. Knockdown of PCAT1 sensitizes MM cells to bortezomib treatment [36]. ANRIL is upregulated in MM patients and correlates to poor prognosis. In vitro and in vivo studies displayed that ANRIL overexpression sustains MM cell growth and resistance to bortezomib-induced apoptosis by interacting with the histone-methyltransferase enzyme EZH2 that epigenetically silences PTEN and mediates chemoresistance [124]. Finally, Hu et al. [27] demonstrated that MALAT1 suppression supports DNA damage via PARP1/LIG3 and promotes cell apoptosis. Simultaneous treatment of MM cells with MALAT-1 oligonucleotides and bortezomib significantly upregulates γH2A.X, a marker of double-strand DNA breaks, suggesting the synergistic effect of anti-MALAT1 and bortezomib treatment. In line with these observations, MALAT1 expression is significantly upregulated in bortezomib-, melphalan- and doxorubicin-resistant MM cell lines. Its inhibition sensitizes resistant MM cells suggesting the pivotal role of MALAT1 in drug resistance [27].

Other studies documented the role of lncRNAs HOTAIR, NEAT1, and CRNDE in resistance to dexamethasone [28,109,127]. Overexpression of HOTAIR in MM patients sustains MM cells viability through the activation of JAK2/STAT3 signaling [125]. NEAT1 is upregulated in dexamethasone-resistant MM cells and prevents MM cells apoptosis by sponging miR-193a and by increasing the expression of MCL-1 protein [107]. Furthermore, NEAT1 may also mediates resistance to melphalan, bortezomib and carfilzomib [28]. CRISPR-mediated deletion of CRNDE reduces MM cell growth and adhesion and increases sensitivity to dexamethasone. CRNDE activates IL-6/IL-6R signaling that promotes MM cells proliferation and resistance to dexamethasone-induced apoptosis [38]. Finally, linc00515 is overexpressed in melphalan-resistant MM cells and regulates chemoresistance by upregulating miR-140-5p and decreasing autophagy via the downregulation of ATG14 [127].

Overall, the evidence suggests that targeting of lncRNAs may represent a new strategy for the treatment of drug resistance in MM.

### 4.3. LncRNAs as Therapeutic Targets: Exploring New Therapies

The key role of aberrantly expressed lncRNAs in the regulation of the “hallmarks of cancer” and in conferring drug resistance makes them promising candidates for therapeutic intervention (Table 2).

Both small interfering RNAs (siRNAs) and shRNAs, initially adopted for mRNA and miRNA inhibition, have been used to therapeutically target lncRNAs. For example, in vitro downregulation of HOXC-AS3 with a specific siRNA induces osteogenic differentiation of the MM-MSCs [84]. Moreover, systemic administration of the in vivo grade HOXC-AS3 siRNA decreases bone lesions in a xenograft MM mouse model, suggesting its potential use in combination therapies [84].

Even though siRNA and shRNA-based approaches have extensively been used for targeting of cytoplasmic RNAs and have been applied for the treatment of several diseases [129], they do not efficiently enter nuclei where the bulk of lncRNAs is located, thus their inhibitory effect on lncRNAs could be limited.

In this regard, a new class of synthetic single-stranded DNA derivatives has been developed. Due to their chemical modification, these antisense oligonucleotides (ASOs) display increased stability and permeability. They specifically bind lncRNAs and induce their degradation by RNAse H [129]. In MM, some studies have investigated the therapeutic potential of lncRNA-specific ASOs both in vitro and in vivo. Two independent studies have shown that the inhibition of MALAT1 expression by different classes of ASOs affects different MM survival pathways and induces tumor cell apoptosis [27,128]. Specifically, Amodio et al. [128] showed that MALAT1 downregulation by a specific ASO restrains proteasome gene expression and promotes MM cell death. Using a different MALAT1-targeting ASO, Hu et al. [27] demonstrated that low MALAT1 promotes DNA damage and apoptosis of tumor cells both in vitro and in a xenograft MM model. Interestingly, these authors used a single wall carbon nanotube (SWCNT) delivery system that protects oligonucleotides from the activity of nucleases and increases their bioavailability without toxicity, making them ideal candidates for potential clinical applications [27,130]. In addition, MM cells treated with a *NEAT1*-specific ASO increase DNA damage and apoptosis [28]. These ASOs-based studies have also investigated the combination of lncRNAs targeting with conventional MM therapies and demonstrated that MALAT1 and NEAT1 ASOs have synergistic effect with bortezomib, carfilzomib, melphalan and PARP1 inhibitor [27,28,130], providing the rationale for the use of lncRNA-specific ASOs in combo therapies.

A new approach for lncRNAs targeting is based on the use of small-molecule inhibitors that can alter the lncRNAs activity through two main mechanisms. First, by binding the docking sites for DNA, RNA, proteins and lipids, they block the interaction of lncRNAs with target elements (as shown for NP-C86 in adipocytes [131]); second, by binding specific structural domains of the lncRNAs, they destabilize/stabilize lncRNAs multidimensional conformation [129]. Even though there is no evidence yet for the use of these compounds in MM, two independent studies have shown that small molecule inhibitors can be used to target MALAT1. In detail, high throughput technologies combined with computational analysis were used to develop specific compounds that target the triple-helix structure MALAT1 [132,133]. The identified molecules interfered with MALAT1 levels and with the expression of its targets in organoids, providing the rationale for their use also in MM models.

Genomic editing system represents an additional method for the specific targeting of lncRNAs, and it has also been applied to MM in vitro models. CRISPR-Cas9 was used to delete a 5.1 kb region spanning exons 4–6 of the CRNDE gene on chromosome 16q12.2. CRNDE downregulation reduces MM cells adhesiveness, IL-6 production by MSCs co-cultured with CRNDE^Δ/Δ^ tumor cells and increases sensitivity to bortezomib [38]. Although these data provide the proof of concept that lncRNA expression can be efficiently targeted using CRSPR-Cas9, the clinical application of this technology is still in its infancy and has only been used for ex vivo genomic manipulations [129].

## 5. Limitations and Future Perspectives

In this manuscript, we have reviewed the influence of lncRNAs on the “hallmarks of cancer” in MM (Figure 2), with a specific focus on their biological and clinical impact, including patient stratification, drug resistance and their potential therapeutic significance.

Despite being a great promise for multiple applications in the oncology field, lncRNAs have still some limitations that need to be addressed.

In particular, lncRNAs are expressed at low levels compared to other ncRNAs (e.g., miRNAs) or mRNAs, making them hard to identify and quantify using the canonical transcriptomic approaches (e.g., microarrays). The advent of high throughput screenings with increased resolution has expanded the number of newly identified lncRNAs (deposited in the LNCipedia v3.1 database, www.lncipedia.org (accessed on 8 January 2019)) that can now be detected in biological fluids such as peripheral blood, offering potential less invasive alternatives for monitoring disease status. Nevertheless, more sensitive detection methods need to be further validated for their application in diagnostic practice.

In addition, as lncRNAs form an intricate network with proteins, mRNAs and other ncRNAs that can affect multiple cellular processes, understanding the contribution of each specific lncRNAs to cellular functions is complex. In silico integrated network analysis could provide a useful tool for clinical stratification of patients. Moreover, the identification of coding/non-coding interactomes would allow the design of additional compounds that alter the lncRNAs activity for therapeutic purposes.

Finally, lncRNAs are differently located in the genome (Figure 1), and the increased expression of specific coding genes could trigger the regulation of different subsets of lncRNAs affecting the expression of other downstream coding/non-coding molecules (miRNA and coding genes). Consequently, the very well-known mutational landscape of tumors should be reconsidered because known mutations could affect not only coding genes, but also the complex non-coding transcriptome network.

These observations suggest that lncRNAs, with their multifaceted biological effects, play a critical role in the clinical heterogeneity of MM and may represent promising targets for MM therapy.

## Figures and Tables

**Figure 1 cancers-14-01963-f001:**
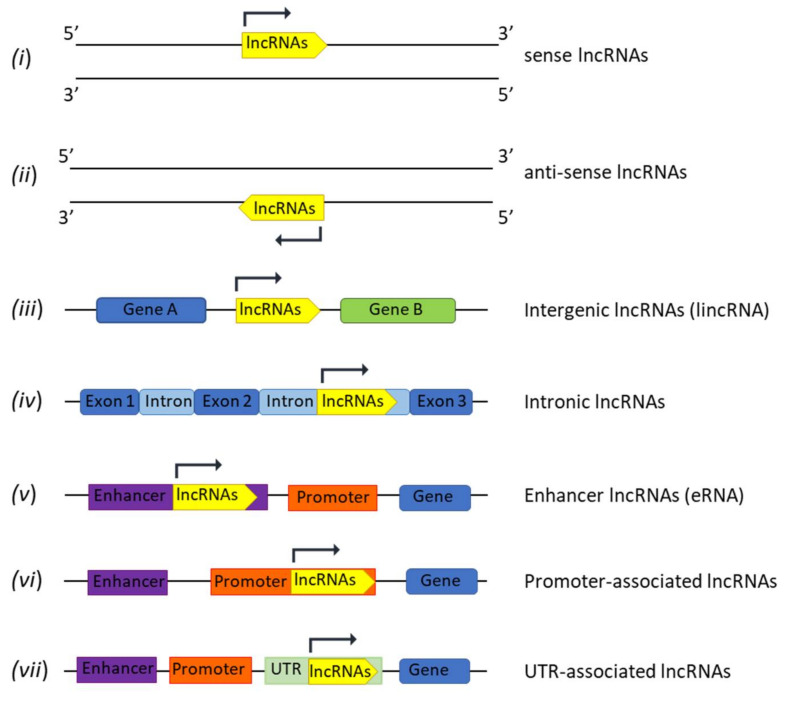
LncRNAs nomenclature. LncRNAs classification based on their orientation and location in the genomic loci.

**Figure 2 cancers-14-01963-f002:**
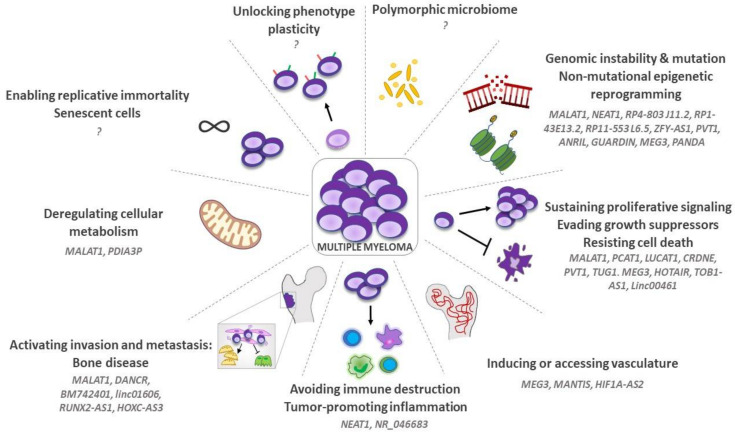
Influence of lncRNA on multiple myeloma “Hallmarks of Cancer”.

**Table 1 cancers-14-01963-t001:** LncRNA expression and clinical correlations in MM.

LncRNA	Expression	Clinical Correlation	Ref.
MALAT1	Upregulated	PFS (shorter)	[32]
NEAT1	Upregulated	OS (shorter)	[107]
CRNDE	Upregulated	OS (shorter)	[38]
PVT1	Upregulated	PFS (shorter)	[108]
UCA1	Upregulated	OS (shorter)	[109]
LINC00461	Upregulated	OS (shorter)	[45]
TCF7	Upregulated	OS (shorter)	[110]
MEG3	Downregulated	OS (shorter)	[43]
OIP-AS1	Downregulated	OS (shorter)	[111]
AC005307.4 AC005307.1 AC005616.1, RP11-161M6.2, RP11-23P13.6, AC005616.1, MIAT	Downregulated	t(11;14)	[112]
RP11-343J3.2, RP11-17M16.2, LINC01102, RP11-345J18.2, ST8SIA6-AS1	Upregulated: RP11-343J3.2, RP11-17M16.2, LINC01102, RP11-345J18.2Downregulated: ST8SIA6-AS1	t(4;14)	[112]
RP11-1085N6.5, RP5-887A10.1, RP11-212I21.4, MIR222HG, LINC00158	Upregulated: RP11-1085N6.5, RP11-212I21.4, LINC00158 Downregulated: RP5-887A10.1, MIR222HG	MAF translocations	[112]
PDLIM1P4, ENSG00000249988, ENSG00000254343	Upregulated	PFS (shorter)	[10]
PDLIM1P4, SMILO, ENSG00000249988	Upregulated: PDLIM1P4 Downregulated: SMILO, ENSG00000249988	OS (shorter)	[10]
RP4-803J11.2, RP1-43E13.2, ZFY-AS1, RP11-553 L6.5	Upregulated in high-risk patients: RP4-803 J11.2, RP1-43E13.2 Upregulated in low-risk patients: ZFY-AS1, RP11-553 L6.5	Survival (shorter in high-risk, longer in low-risk)	[29]

PFS: Progression-Free Survival, OS: Overall Survival.

**Table 2 cancers-14-01963-t002:** LncRNAs as therapeutic targets in MM.

LncRNA Target	Method	Platform	Effect	Ref.
HOXC-AS3	siRNA	In vitro	MM-MSCs osteogenic differentiation	[84]
HOXC-AS3	In vivo grade siRNA	In vivo	Decreased bone loss	[84]
MALAT1	ASO	In vitro	Reduced tumor cell viability and motility	[128]
MALAT1	ASO	In vivo	Reduced tumor growth	[128]
MALAT1	ASO	In vitro	Reduced tumor cell viability, increased sensitivity to Bortezomib, Melphalan, Doxorubicin	[27]
MALAT1	SWCNT-ASO	In vivo	Reduced tumor growth, increased survival	[27]
NEAT1	ASO	In vitro	Reduced tumor cell viability, increased sensitivity to Bortezomib, Melphalan, Carfilzomib	[28]
NEAT1	ASO	In vivo	Reduced tumor growth	[28]
CRNDE	CRISPR-Cas9	In vitro	Reduced IL6R expression, Reduced tumor cell proliferation, increased sensitivity to Bortezomib, Dexamethasone	[38]
CRNDE	CRISPR-Cas9	In vivo	Reduced tumor growth	[38]

siRNA: small interfering RNA, ASO: antisense oligonucleotide, SWCNT: single wall carbon nanotube.

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
