# Peer review of "The Landscape of lncRNAs in Multiple Myeloma: Implications in the “Hallmarks of Cancer”, Clinical Perspectives and Therapeutic Opportunities"

_cancers, 2022, doi:10.3390/cancers14081963_

Round 1

Reviewer 1 Report

This review by Saltarella’s group is focused on the role of lncRNAs in the Hallmarks of multiple myeloma, and their implication in the stratification of patients and their use as potential therapeutic targets.

The review is overall well-written, but the authors should be careful with minor typos:

  • letter size in page 8
  • justification effect in page 12
  • Figure 2: Text says “enabling relicative"

Major comments:

  • The division of the paragraphs is confusing. I recommend separating the point 1.1 from the rest of the point 1, focusing on a “new” point 2 for only all the hallmarks of cancer. Besides, the order of the hallmarks should be the same in the text and in the figure, mixing them is also confusing for readers.

  • The tittle of the review specifies about “clinical perspectives and therapeutic opportunities”, but I noticed more information and examples were missing about these topics. Authors should explain more these topics.

  • Authors should review the latest published articles in order to include novel information about lncRNAs that has not yet been reviewed. There is lack of information about recently published articles about lncRNAs such as LINC01473 (Peng F, et al 2022), ANRIL (Yang LH, et al 2021), lnc-TCF7 (Liu H, et al 2021),…

Author Response

POINT-BY-POINT ANSWERS TO REVIEWER COMMENTS

 The authors thank the Reviewer for helpful criticism and are glad for positive comments.

 Reviewer’s comments: This review by Saltarella’s group is focused on the role of lncRNAs in the Hallmarks of multiple myeloma, and their implication in the stratification of patients and their use as potential therapeutic targets.

The review is overall well-written, but the authors should be careful with minor typos:

  • letter size in page 8
  • justification effect in page 12
  • Figure 2: Text says “enabling relicative"

Reply: We thank the Reviewer for his comments. Accordingly, typos and formatting have been corrected.

Reviewer’s comment: The division of the paragraphs is confusing. I recommend separating the point 1.1 from the rest of the point 1, focusing on a “new” point 2 for only all the hallmarks of cancer. Besides, the order of the hallmarks should be the same in the text and in the figure, mixing them is also confusing for readers.

Reply: Accordingly to Reviewer comment, we have re-numbered the paragraphs and modified the figure, so the order of the hallmarks matches the text.

Reviewer’s comment: The tittle of the review specifies about “clinical perspectives and therapeutic opportunities”, but I noticed more information and examples were missing about these topics. Authors should explain more these topics.

Reply:  We thank the Reviewer for his comments. Accordingly, we have modified the title of paragraph 4 as follow “Clinical perspective and therapeutic opportunities of lncRNAs” and we added some sentences explaining in detail clinical involvement of lncRNAs:

  • Page 10 lines 375-377: “Zhou et al. [25] developed a lncRNA-focused risk model for survival prediction based on the expression of 4 lncRNAs (RP4-803 J11.2, RP1-43E13.2, ZFY-AS1 and RP11-553 L6.5) that regulate processes such as cell proliferation and RNA repair[25].”
  • Page 10 lines 387-390: “For example, lncMCL1-2 expression is negatively correlated to its targets mir106a-5p, miR18a-5p, miR18b-5p and miR17-5p that modulate MCL1 expression, suggesting that the miRNA sponging activity of lncMCL1-2 can directly affect survival of MM cells [115].”
  • Page 10 lines 391-392: “They identified 39 lncRNAs and 1445 mRNAs involved in several biological processes with a direct impact on MM clinical course [116].”

Reviewer’s comment: Authors should review the latest published articles in order to include novel information about lncRNAs that has not yet been reviewed. There is lack of information about recently published articles about lncRNAs such as LINC01473 (Peng F, et al 2022), ANRIL (Yang LH, et al 2021), lnc-TCF7 (Liu H, et al 2021),…

Reply: We thank the reviewer for this comment, accordingly, we added and commented in main text the 3 suggested papers. In particular:

  • Peng F et al. 2022 – Page 8 lines 295-299: “Finally, a recent study has demonstrated that lncRNAs are also altered in MM osteoblasts compared to healthy controls. The analysis by Peng and colleagues [85] has revealed that MM osteoblasts display an upregulated expression of the new lncRNA LINC01473 together with CD74 mRNA. Although the functional consequences of this altered profile have not been investigated yet, authors suggested their possible involvement in OBD and immune escape [85].”
  • Liu H et al. 2021 – Page 10 lines 363-366: Increased expression of several lncRNAs, such as MALAT1 [28], NEAT1 [104,105], CRNDE [34], PVT1 [106], TUG [38], UCA1 [107], LINC00461 [41], and TCF7 [108], has been correlated to shorter PFS, and downregulated expression of MEG3 [39] and OIP5-AS1 [109] has been associated to worse prognosis.
  • Yang LH et al. 2021 - Page 11 lines 415-418: “ANRIL is upregulated in MM patients and correlates to poor prognosis. In vitro and in vivo studies displayed that ANRIL overexpression sustains MM cell growth and resistance to bortezomib-induced apoptosis by interacting with the histone-methyltransferase enzyme EZH2 that epigenetically silences PTEN and mediates chemoresistance [122].”

Because the new References 85, 108 and 122 have been inserted, all the other following references have been re-numbered accordingly.

Reviewer 2 Report

In this review entitled “The Landscape of lncRNAs in Multiple Myeloma: Implications in the “Hallmarks of Cancer”, Clinical Perspectives and Therapeutic Opportunities” the authors provide a detailed description on the accumulated data regarding the role of lncRNAs in the “hallmarks of cancer”, mostly focusing on the different intracellular pathways that lncRNAs are involved in the multi-step process of multiple myeloma (MM) cancer development. They also summarize the existing knowledge with up-to-date studies concerning the clinical significance and therapeutic opportunities of lncRNAs in MM.

This is a well written and very interesting review article, summing up the majority of substantial studies regarding lncRNAs functional role and clinical value in MM. However there are a few minor revisions that the authors have to address.

  1. Introduction: A brief paragraph about lncRNAs’ role in cancer should be included.
  2. The section “lncRNAs biogenesis and functions” as well as Figure 1 with lncRNAs nomenclature in the revised version of the manuscript is missing.
  3. lncRNAs functions can be divided into different groups depending on their molecular mechanisms (signal, decoy, scaffold or guide lncRNAs). Please discuss accordingly.
  4. In the section “Clinical perspectives and therapeutic opportunities of lncRNA” a table that summarizes the clinical value of lncRNAs in MM would be useful.

Reviewer 3 Report

The authors review the effects of IncRNAs in myeloma and focus on clinical perspectives and therapeutic applications. 

  1. There are missing paragraph on section 2, entitled" IncRNAs biogenesis and functions."
  2. It is convenience to have the summery of trials in tables.

Round 2

Reviewer 1 Report

After reading the revised version and the cover letter, I would just suggest being careful with some typos, like double spaces (lines 52, 71, 155, 261...).
